# Formaldehyde reacts with N-terminal proline residues to give bicyclic aminals

Tobias John[1], Elisabete Pires[1], Svenja S. Hester[2], Eidarus Salah[1], Richard J. Hopkinson [3✉] & Christopher J. Schofield [1✉]

Formaldehyde (HCHO) is a potent electrophile that is toxic above threshold levels, but which is also produced in the nuclei of eukaryotic cells by demethylases. We report studies with the four canonical human histones revealing that histone H2B reacts with HCHO, including as generated by a histone demethylase, to give a stable product. NMR studies show that HCHO reacts with the N-terminal proline and associated amide of H2B to give a 5,5-bicyclic aminal that is relatively stable to competition with HCHO scavengers. While the roles of histone modification by this reaction require further investigation, we demonstrated the potential of N-terminal aminal formation to modulate protein function by conducting biochemical and cellular studies on the effects of HCHO on catalysis by 4-oxalocrotonate tautomerase, which employs a nucleophilic N-terminal proline. The results suggest that reactions of N-terminal residues with HCHO and other aldehydes have potential to alter protein function.

[1] Chemistry Research Laboratory, 12 Mansfield Road, Oxford OX1 3TA, UK. [2] Nuffield Department of Medicine, Target Discovery Institute, University of Oxford, Oxford, UK. [3] Leicester Institute for Structural and Chemical Biology and School of Chemistry, University of Leicester, Henry Wellcome Building, Lancaster Road, Leicester LE1 7RH, UK. ✉email: richard.hopkinson@leicester.ac.uk; christopher.schofield@chem.ox.ac.uk

As a consequence of its highly electrophilic nature and aqueous solubility, formaldehyde (HCHO) reacts readily with nucleophilic biomedicinally relevant macromolecules, including proteins, nucleic acids, and sugars, as well as small molecules such as amino acids, peptides and many drugs[1–5]. HCHO is toxic at high concentrations; however, the chemical details underlying this toxicity, which likely involves chromatin modifications, are unclear[6–8]. The major HCHO detoxification pathway in animals is proposed to involve reaction with glutathione (GSH), followed by subsequent oxidation to give formate[9]. Humans are exposed to exogenous HCHO from sources including drugs[10], cosmetics[11], smoke[12], and paper[13]. However, HCHO is also a human metabolite produced by N-methyl group enzymatic demethylation[14] and during serine metabolism[15]. It is thus possible that, in addition to being a toxin, HCHO and HCHO-derived metabolites/adducts have functional roles[16].

N-Methylation and demethylation of lysine residues in histone tails play important roles in the regulation of eukaryotic transcription[17]. Pioneering work demonstrated that HCHO is a coproduct during demethylation of $N^\varepsilon$-methylated lysine-residues on histones[18]. Subsequent studies showed oxidative $N^\varepsilon$-methyl-lysine residue demethylations are catalysed by the flavin-dependent lysine specific demethylases and the Jumonji C demethylases (JmjC KDMs); the activities of both demethylase classes produce HCHO as a co-product[14,19]. Given that histone tails contain multiple basic/nucleophilic residues with the potential to react with HCHO and other reactive electrophiles[17], we are interested in investigating the nature of reactions with them, including with respect to the potential for HCHO-mediated regulation and scavenging.

We now report biochemical studies, which reveal that histone H2B is unusually susceptible to reaction with HCHO, forming a relatively stable 5,5-bicyclic aminal structure that is derived by reaction of HCHO with the H2B N-terminal proline residue and the associated amide group. The selectivity of the reaction was investigated using proline analogues and other aldehydes. Studies using 4-oxalocrotonate tautomerase (4-OT) as a model enzyme reveal that HCHO can directly inhibit activity by reacting with an N-terminal proline involved in catalysis, including in cells. Given multiple proteins have N-terminal prolines, including >200 predicted human proteins[20], there is substantial potential for regulation of protein function and stability by reaction of N-terminal proline residues with HCHO or other reactive aldehydes.

## Results and discussion

**HCHO reacts with histone H2B to form a +12 Da adduct.** Our initial studies focused on investigating the reactions of histone tail fragments with HCHO. Peptides containing the first 15 residues of the canonical histones (H2A, H2B, H3, and H4) were mixed with different HCHO concentrations (equimolar to $10^4$-fold excess) at pH 7.4, then analysed by mass spectrometry (MS) over 24 hours. All the peptides reacted with HCHO, with mass shifts of +12 Da, +24 Da and +36 Da (relative to starting peptide) being observed, implying formation of multiple methylene group adducts (Figs. 1 and S1). No evidence for intermolecular peptide cross-linking was observed in these studies.

Notably, the H2A, H3 and H4 peptides only reacted substantially when mixed with a $10^4$-fold excess of HCHO, whereas the H2B peptide was observed to react at lower HCHO concentrations and, at least predominantly, to give a single apparent +12 Da mass shift (Figs. 1b and S2). The extent of +12 Da adduct(s) formation was increased at 37 °C (Fig. 1c) and under mildly alkaline conditions (pH 8.2, Fig. 1d). MS fragmentation studies supported the proposal that a single (major) stable +12 Da adduct is formed and that the modification

involves the H2B N-terminal proline residue (Fig. 1e, f). This assignment was supported by MS analyses with an H2B P1A variant peptide, which reacted much less efficiently with HCHO (Fig. S3d). MS fragmentation studies conducted on the product of the reaction of recombinant H2B protein with a 100-fold excess of HCHO revealed the +12 Da modification occurs at the N-terminus of H2B (Fig. S4).

To investigate whether KDM catalysis can produce sufficient HCHO to modify H2B, analyses were then conducted on samples containing the 15 residue H2B peptide, the human JmjC KDM KDM4E, and its histone H3 peptide substrate ($H3K9Me_3$). Analysis of the $H3K9Me_3$ peptide in the mixture revealed KDM4E-dependent demethylation, as evidenced by 14 Da mass decreases of the $H3K9Me_3$ peptide. The H2B peptide in the mixture reacted to give a product with a +12 Da mass increment (Figs. 2a, b and S5b, c). Addition of N-oxalylglycine, a JmjC KDM inhibitor[21], abolished both $H3K9Me_3$ demethylation and formation of the +12 Da H2B adduct (Fig. 2c, d), showing the potential for JmjC KDM/HCHO-enabled H2B modification.

**HCHO forms a 5,5-bicyclic aminal with N-terminal proline residues.** Studies then focused on assigning the structure of the +12 Da H2B adduct, initially by NMR analyses with a shorter H2B peptide (NH-PEPAK-$NH_2$) and $^{13}C$-labelled HCHO (Fig. S3a–c). Compared to the starting peptide, chemical shift changes in the $^1H$ resonances corresponding to the N-terminal prolyl α, γ, and δ hydrogens were observed (Fig. S3a). Two new $^1H$ resonances were assigned to the hydrogens of a methylene bridge ($\delta_H$ 4.3 ppm and $\delta_H$ 4.5 ppm, Fig. S3b). HMBC correlations from $^1H$ resonances at $\delta_H$ 2.59 ppm and $\delta_H$ 3.14 ppm, corresponding to the N-terminal prolyl δ hydrogens, to the methylene bridge $^{13}C$ resonance (at $\delta_C$ 66 ppm), imply modification on the N-terminal prolyl secondary amine (Fig. S3b). Studies on the reaction of an N-terminal H2B dipeptide (ProGlu-OH, **2**) with HCHO (10-fold) at pH 10 revealed formation of an analogous adduct that could be isolated by HPLC. NMR analyses of this isolated product revealed that the methylene bridge is connected via the prolyl amine and the nitrogen of the prolyl-glutamate peptide bond, revealing formation of a 5,5-bicyclic aminal (Figs. 3a and S6). This assignment was supported by reductive methylation of **2** using HCHO and $NaCNBH_3$; this reaction gave the α−amino methylated proline product, consistent with methylene bridge formation via an N-terminal iminium ion (Fig. S7). The formation of the 5,5-bicyclic aminal **2a** has precedent in the formation of bicyclic lactones formed by reaction of proline and pivalaldehyde, as pioneered by Seebach et al.[22,23].

The selectivity of bicyclic aminal formation was then investigated. Dipeptide analogues of **2**, in which the N-terminal proline was replaced with azetidine- (AzeGlu, **1**), piperidin-e (PipGlu, **3**) or alanine- (AlaGlu, **4**) containing analogues, were treated with HCHO. Product structures were investigated by NMR after HPLC purification where possible. Formation of apparent bicyclic adducts were observed with the piperidine dipeptide **3** (**3a**, Figs. 3a and S8) and azetidine dipeptide **1** (**1a**, Figs. 3a and S9); however, the likely 4,5-azetidine bicycle derived from **1a** was unstable, apparently degrading to form an N-terminal hemiaminal (compound **1b**, Figs. 3a and S10). Alanine dipeptide **4** reacted to form a product containing two HCHO-derived methylene groups, that is a monocyclic aminal and a hemiaminal linked to the N-terminal nitrogen (compound **4a**, Figs. 3b and S11).

To investigate the stability and reversibility of the HCHO adducts (**1a + b, 2a, 3a, 4a**), the dipeptides were reacted with a 10-fold excess of HCHO at pD 10, then treated with the HCHO scavenger 1,3-cyclohexanedione (1,3-CHD)[14]. Dipeptides **2a** and

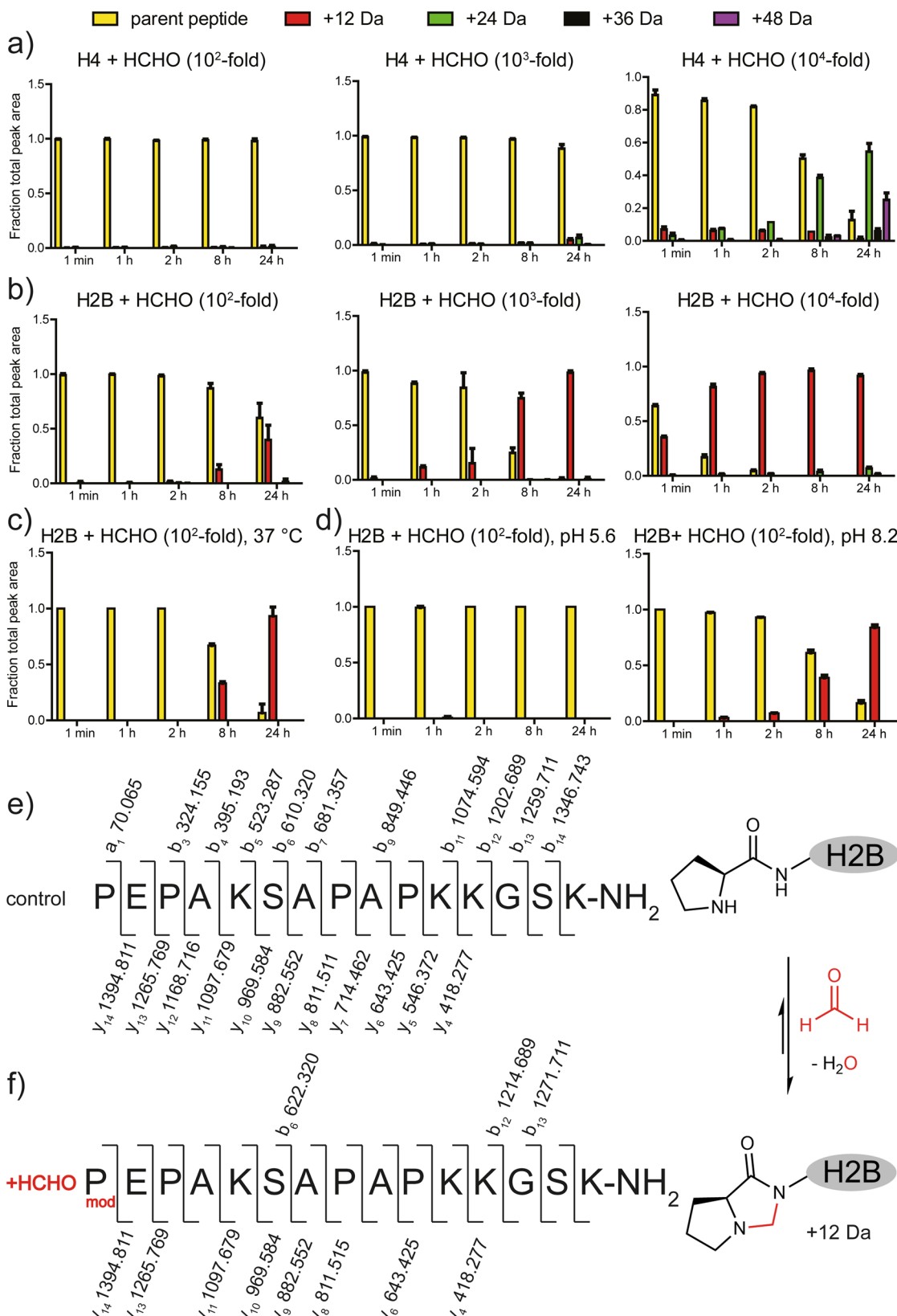

**3a** were relatively stable over the analysis period (22 h, Figs. 3a and S12–S14); however, **1a + b** underwent 1,3-CHD-enabled reaction to give **1** (Figs. 3a and S15). The monocyclic methylene bridge of **4a** was stable over the analysis period, but the hemiaminal group of **4a** was lost to give **4b** (Figs. 3b and S16). 1,3-CHD treatment of the H2B pentapeptide-derived HCHO

adduct revealed no apparent degradation of its methylene bridge (Fig. S17a, panel 1). Compound **2a** was also apparently stable in the presence of the biologically relevant HCHO scavengers cysteine and GSH[4,9] and across a biologically relevant pH range, even with the scavengers in 40-fold excess (Fig. S17a, panels 3–5; Fig. S17a, panels 6–8).

**Fig. 1 The N-terminus of histone H2B reacts efficiently with HCHO to form a product with a +12 Da mass increment. a** Reaction of the H4 15mer peptide ($NH_2$-SGRGKGGKGLGKGGA-$NH_2$, 2.5 µM) with HCHO (250 µM, 2.5 mM or 25 mM) at pH 7.4 and ambient temperature. **b** Reaction of the H2B 15mer peptide (NH-PEPAKSAPAPKKGSK-$NH_2$, 2.5 µM) with HCHO (250 µM, 2.5 mM or 25 mM) at pH 7.4 and ambient temperature. Formation of a +12 Da adduct is observed. **c** Reaction of the H2B 15mer peptide (2.5 µM) with a 100-fold excess of HCHO (250 µM) at 37 °C. Formation of a +12 Da species is favoured. **d** Reactions (ambient temperature) of the H2B 15mer peptide (2.5 µM) with a 100-fold excess of HCHO (250 µM) at pH 5.6 (left) and 8.2 (right). Formation of the +12 Da species is favoured at pH 8.2. HCHO adducts are in: red (+12 Da), green (+24 Da), black (+36 Da) and purple (+48 Da). The parent peptide (unreacted) is in yellow. Reactions for **a–d** were monitored by MALDI MS after HCHO addition (1 min), after 1, 2, 8, and 24 h. Errors: SD of the mean ($n = 3$, technical repeats). MALDI MS/MS results summary with H2B 15mer peptide without (**e**) and with (**f**) HCHO addition. MS/MS analyses show the H2B peptide (C-terminal amide) and the generated a-, b- (N-terminal) and y-ions (C-terminal) obtained after fragmentation, indicating that the +12 Da shift occurs at the N-terminus.

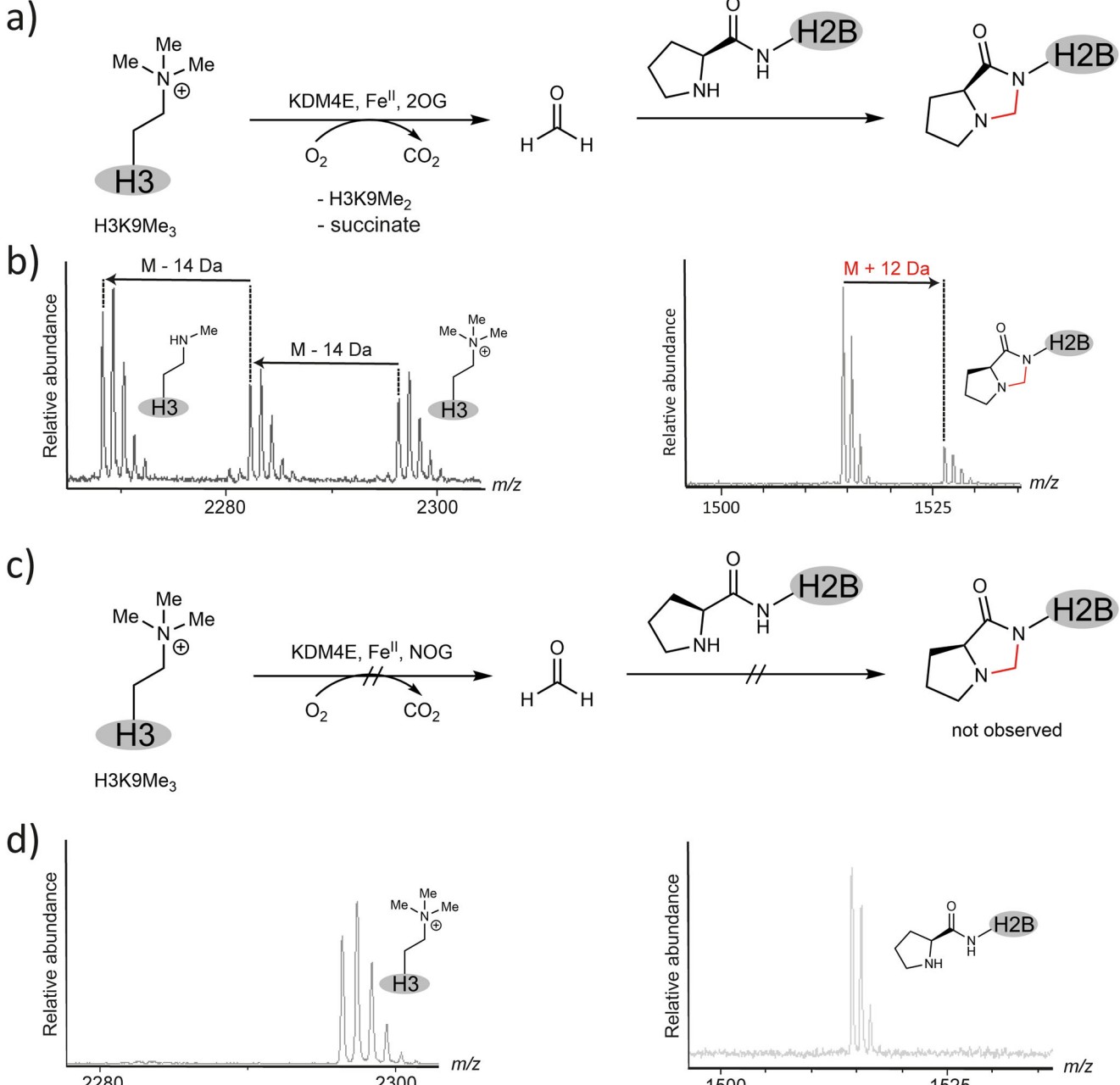

**Fig. 2 In situ-generated HCHO by a demethylase reacts with H2B to form a +12 Da adduct. a** Formation of a +12 Da adduct derived from the H2B N-terminal proline and HCHO produced during KDM4E-catalysed $N^ε$-methyllysine demethylation. MALDI MS spectra showing demethylation of H3K9Me3 21mer peptide (2296 Da) to form dimethylated (2282 Da, M-14 Da) and monomethylated (2268 Da, M-28 Da) products in the presence of 2-oxoglutarate (2OG) and KDM4E (24 h incubation) (**b**, left panel), indicative of HCHO formation. With H2B in the reaction mixture, the H2B 15mer peptide reacts with the KDM4E-generated HCHO (24 h) to form a +12 Da HCHO-derived adduct in the presence of 2OG, the co-substrate of KDM4E (**b**, right panel). $Na^+$ adducts of the parent peptide [1514 Da] and HCHO-adduct [1526 Da, M + 12 Da] are observed. **c, d** In the presence of N-oxalylglycine (NOG), a broad-spectrum 2OG-oxygenase inhibitor, demethylation is inhibited and less HCHO is formed and no reaction with H2B is observed (**d**, right panel).

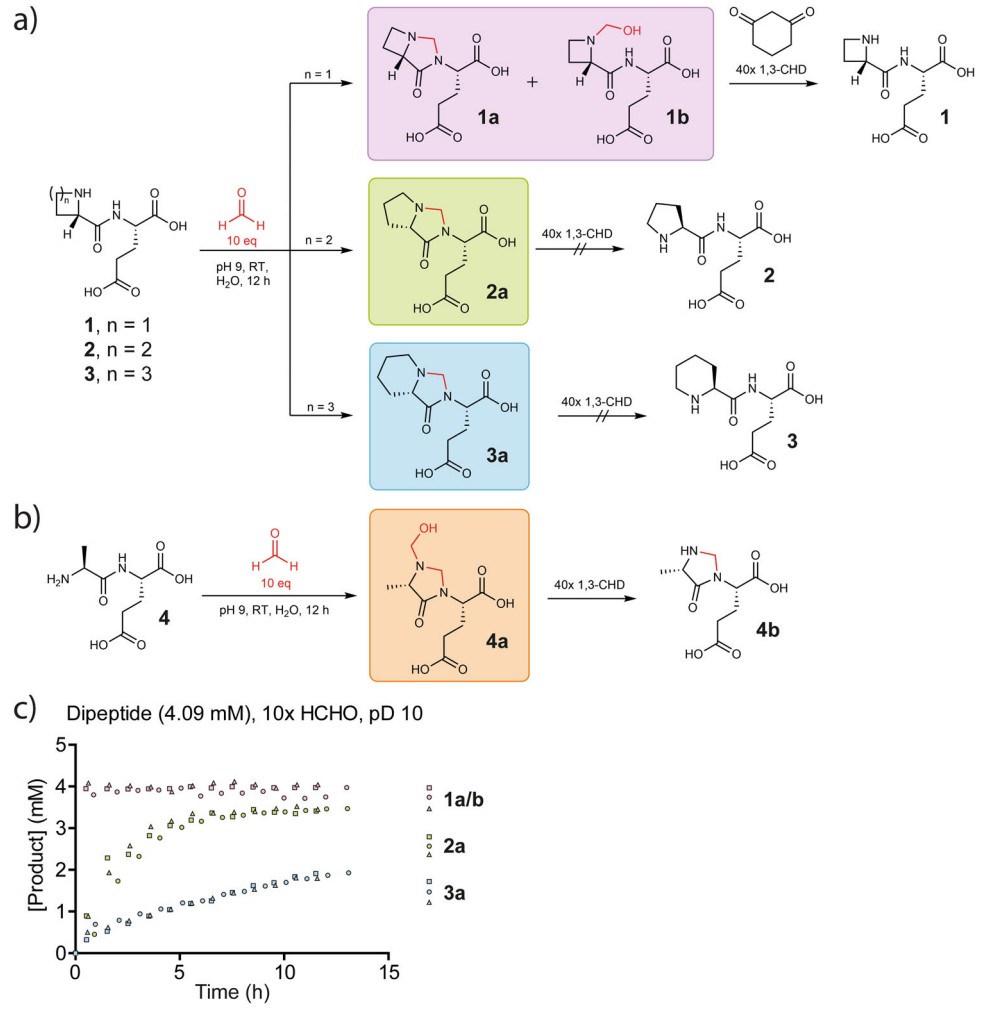

**Fig. 3 HCHO reacts with an N-terminal proline to form a methylene bridge between an N-terminal nitrogen and the adjacent peptide bond nitrogen.**
**a** Products of reactions between HCHO and the dipeptides AzeGlu ($n = 1$, **1**), ProGlu ($n = 2$, **2**), and PipGlu ($n = 3$, **3**). A 10-fold molar excess of aqueous HCHO (410 µmol) was added to each dipeptide (41 µmol). Products containing apparent methylene bridges were observed with all peptides, although the methylene adduct formed with **1** (**1a**) degraded after addition of the HCHO scavenger 1,3-cyclohexandione (1,3-CHD, 40-fold excess). **b** AlaGlu (**4**, 41 µmol) reacts with a 10-fold excess of HCHO (410 µmol) at pD 12 and ambient temperature to form the methylene bridge-containing adduct **4a**. Addition of 1,3-CHD (40-fold excess) results in loss of the HCHO-derived hemiaminal from **4a**, forming **4b**. **c** Graph showing NMR time-courses monitoring reaction of a 10-fold HCHO excess with dipeptides **1–3** at pD 10 ($n = 3$ independent replicates); **4** did not react to give an observed product under these conditions (Fig. S18d).

The reactions of the dipeptides with HCHO (10-fold excess, pD 10) were studied in real time by [1]H NMR (Figs. 3c and S18). **1** Reacted most efficiently; the reaction was too fast to determine the initial reaction rate, while the formation rate of **1a** was >1.73 µM s$^{-1}$. Dipeptide **2** was slower to react (**2a** formation rate 0.24 µM s$^{-1}$), while **3** was the slowest to react (**3a** formation rate 0.054 µM s$^{-1}$). Dipeptide **4** apparently did not react to give a detected product under these conditions (Fig. S18d).

To investigate the selectivity of **2**, and by extension H2B, for reaction with HCHO, mixtures were prepared containing **2** and other biologically relevant carbonyl compounds (pyruvate, 2OG, acetone, glyoxylic acid and acetaldehyde, 100-fold excess). After incubation for 12 hours at 37 °C, LC/MS analyses revealed no adduct formation in the samples with pyruvate, 2OG, acetone and glyoxylic acid; however, some evidence for an adduct was observed with acetaldehyde (Fig. S19a, c). LC/MS analyses suggest that the acetaldehyde-derived adduct degrades during isolation (Fig. S19b).

We then investigated (1) whether **2** can remove HCHO-derived methylene bridges/adducts from thioproline, metampicillin or S-(hydroxymethyl)glutathione (HMG)[9], and (2) whether

**2** can compete with cysteine or glutathione (GSH) for reaction with HCHO. Addition of **2** to samples of pre-formed thioproline, metampicillin or HMG did not reveal formation of bicyclic aminal **2a** (Figs. S20a–c and S21). Competition experiments involving reaction of mixtures of **2** and cysteine or GSH with HCHO (added last) revealed no significant formation of **2a**, though formation of reported cysteine or GSH HCHO-derived adducts was observed (Figs. S20d, e and S22)[4,9]. Collectively, these studies suggest that HCHO and proline adducts as in **2** will not, at least efficiently, sequester HCHO from many reported stable HCHO-derived adducts, or out-compete cysteine or GSH for reaction with HCHO under the tested conditions (Fig. S20).

Preservatives including HCHO-releasing agents are used to hinder microbial growth in cosmetics-related products, but concerns have been raised over their effects on skin and their links to cancer and asthma[24,25]. To investigate whether representative HCHO-releasing agents from cosmetics can induce methylene adduct formation with H2B, the HCHO donors diazolidinyl urea (Fig. S23e) and imidazolidinyl urea (Fig. S23f) were incubated with the H2B 15-residue peptide. The

mixtures were analysed by MALDI MS. A +12 Da shift on the H2B peptide was observed after 8 h incubation with a 100-fold excess of the HCHO donors (Fig. S23a, b); MS analyses confirmed that the modification occurs at the N-terminal proline (Fig. S23c, d). Preliminary analysis suggests diazolidinyl urea is more reactive than imidazolidinyl urea under the tested conditions (Fig. S23a, b), an observation which correlates with their relative HCHO release rates in water[26].

We then investigated whether the H2B-HCHO adduct can be identified in human HEK293T cells treated with HCHO. While HCHO impaired cell viability at concentrations greater than 200 μM (Fig. S24), MS-based analyses (proteomics) did not result in sufficient coverage of the H2B N-terminal region to enable identification of N-terminal adducts (Fig. S25). Given that N-terminal amino acids and their modifications can regulate protein stability by affecting degradation (the N-end rule)[27], we next investigated whether addition of HCHO to cells affects H2B turnover. HEK293T cells were treated with cycloheximide (CHX) to inhibit protein synthesis, with addition of HCHO and protein quantification. HCHO treatment consistently led to apparently higher H2B levels after prolonged CHX treatment (>10 h, Fig. S26a–c), but higher levels of H3 and H2A were also observed (Fig. S26c). RNA-Seq analyses indicated increased levels of H2B mRNA in cells treated with HCHO (0.37-fold less in untreated cells, $q = 0.043$, Fig. S26e), but increases were also observed for both H2A mRNA (0.33-fold less in untreated cells, $q = 0.043$) and H3 mRNA (0.42-fold less in untreated cells, $q = 0.034$). Studies on another protein bearing an N-terminal proline, leukotriene A-4 hydrolase (LTA4H), which is not related to histones, revealed no evidence for a HCHO-dependent effect on its stability (Fig. S26c, d). Therefore, it appears that HCHO may have a limited direct effect in substantially regulating protein stability in cells, at least for H2B and LTA4H in HEK293T cells under our conditions.

**HCHO reacts with 4-OT and inhibits catalysis**. The biochemistry of human histones/chromatin in cells is extremely complex, thus we searched for a simpler model system to investigate the possibility that the reaction of non-cytotoxic levels of HCHO with an N-terminal proline residue can alter protein function. We identified the N-terminal nucleophilic enzyme 4-oxalocrotonate tautomerase (4-OT)[28] as a candidate. 4-OT catalyses the tautomerisation of 4E-2-oxo-hexenedioate (4-oxalocrotonate) to give conjugated 3E-2-oxo-hexenedioate in *Pseudomonas putida*[28]. While 4-OT is structurally and mechanistically unrelated to H2B, it also has an N-terminal proline residue, but one which is employed as an essential nucleophile during 4-OT catalysis. 4-OT activity was investigated by NMR and absorbance-based assays monitoring 4-OT-catalysed Michael addition of acetaldehyde to *trans*-β-nitrostyrene. In these assays, the N-terminal proline is proposed to react with acetaldehyde to form an enamine, which reacts with *trans*-β-nitrostyrene to form 4-nitro-3-phenylbutanal (Fig. 4a)[29,30].

Kinetic parameters for 4-OT-catalysed formation of 4-nitro-3-phenylbutanal were determined using an absorbance assay, which revealed a relatively high $K_M$ value for acetaldehyde ($K_M = 281$ μM ± 42 for *trans*-β-nitrostyrene; $K_M = 74$ mM ± 4.8 for acetaldehyde, Fig. S27). When purified recombinant 4-OT was pre-incubated with a 200-fold excess of HCHO, a substantial reduction in the rate of 4-nitro-3-phenylbutanal formation was observed by NMR (Figs. 4b, c and S28a). Only a small increase in inhibition of 4-OT by HCHO was observed as a function of increasing pre-incubation time, suggesting reaction with HCHO is fast (Fig. S28b, c). With increasing acetaldehyde concentrations, the inhibition mediated by a 200-fold excess of HCHO was less

significant, consistent with competition between acetaldehyde and HCHO (Fig. S28d). Circular dichroism analyses showed no evidence for alterations in 4-OT secondary structure after HCHO treatment in the tested concentrations (Fig. S29b). However, a +12 Da modification on the N-terminal proline of 4-OT was observed by LC-MS/MS fragmentation analysis following trypsin digestion (Fig. S30). Matrix assisted laser desorption ionisation (MALDI) MS data on intact 4-OT showed a time-dependent increase of a +24 Da species, suggesting the potential formation of two methylene adducts (Fig. S29a). Treatment with HCHO followed by reduction with NaCNBH₃ resulted in a +14 Da adduct, providing evidence for formation of an N-terminal N-methyl-proline residue (Fig. S31). The formation of the +14 Da adduct was much faster than the formation of the +12 Da aminal methylene bridge (-N-CH₂-N-) with HCHO only (Fig. 1b). These results demonstrate that HCHO can react reversibly with the N-terminal proline of 4-OT; however, the reaction with HCHO may be more dynamic/complex than with H2B, possibly reflecting the role of the N-terminal proline in 4-OT catalysis. It is possible that 4-OT inhibition in solution is mediated by hemiaminal formation instead of, or in addition to, bicyclic aminal formation.

We then investigated whether HCHO-dependent inhibition of 4-OT occurs in bacterial cells. BL21(DE3) *Escherichia coli* cells producing recombinant 4-OT (Fig. S32a) were treated with *trans*-β-nitrostyrene and acetaldehyde using a reported procedure[31]; the absorbance signal from *trans*-β-nitrostyrene was used to monitor 4-OT catalysis. Efficient *trans*-β-nitrostyrene depletion was observed in the 4-OT-producing cells (Fig. 4d), with only slow depletion in empty vector control cells (the low-level depletion in the controls is likely due to degradation and/or reaction of *trans*-β-nitrostyrene with cellular nucleophiles). HCHO addition was clearly observed to inhibit 4-OT activity in cells in a dose-dependent manner (Fig. 4d, e). The concentrations of HCHO observed to induce 4-OT inhibition either did not inhibit, or only mildly inhibited, cell proliferation (350 μM or 700 μM HCHO respectively, Fig. S32b). In bacteria producing 4-OT, HCHO treatment resulted in formation of a +12 Da modification on the N-terminal proline of 4-OT, as observed by in-gel digestion of 4-OT followed by MS analysis (Figs. S33–S35). Overall, these studies provide evidence that HCHO can inhibit the activity of 4-OT in bacterial cells by reacting with its N-terminal proline.

## Conclusion

Formaldehyde-mediated cross-linking reactions involving proteins and nucleic acids are well known[3,4]. Of particular relevance to our work is cross-linking of the bacterial transcription factor FrmR, which switches on the HCHO detoxification response following HCHO-mediated intramolecular linking of its N-terminal proline and a cysteine residue[32]. Our combined results show that HCHO can also react with an N-terminal proline and its amide bond, though to reversibly form a 5,5-bicyclic aminal **2a**, without involvement of a cysteine residue, under physiologically relevant conditions. Although the interactions between HCHO (or other reactive aldehydes) and nucleosomes are likely complex, the results imply that HCHO reacts more efficiently with the secondary amines of N-terminal prolines than with the primary amines of the N-terminal alanine and serine residues of H2A, H3 and H4 N-termini, at least to give stable bicyclic products.

The other tested carbonyl compounds, including acetaldehyde, did not exhibit similar reactivity; this is likely in part due to steric factors influencing product stability (although reduced reactivity may also play a role). The findings thus suggest that HCHO is an

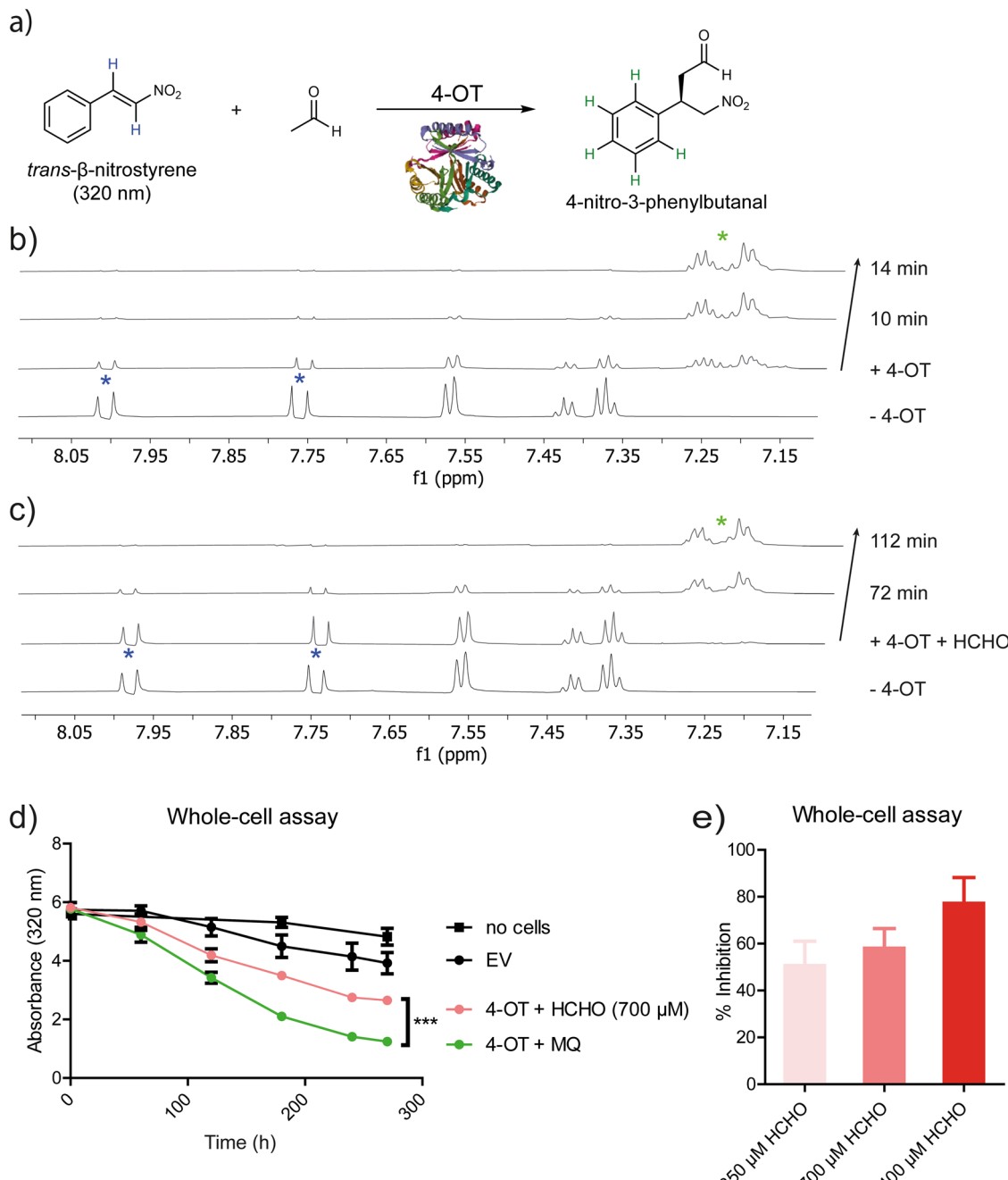

**Fig. 4 HCHO inhibits catalysis by isolated 4-OT and 4-OT in cells. a** 4-OT-catalysed reaction of acetaldehyde with *trans*-β-nitrostyrene to give 4-nitro-3-phenylbutanal. NMR analysis of the alkene hydrogens (blue) and aromatic product hydrogen atoms (green) was used to monitor reaction. Protein image: derived from PDB entry 4X19[46]. **b** $^1$H NMR spectra (700 MHz) showing time-dependent 4-OT-catalysed formation of 4-nitro-3-phenylbutanal (green asterisks). 4-OT was pre-incubated with buffer as a control (17 h, 37 °C). Complete substrate conversion is observed after 14 min. **c** $^1$H NMR spectra showing the effect of HCHO on 4-OT-catalysed formation of 4-nitro-3-phenylbutanal. 4-OT (52.9 μM) was pre-incubated with 200 equivalents of HCHO (10.6 mM) for 17 h at 37 °C. Near-complete substrate conversion is only observed after >72 min. Note, at the first analysed time point with HCHO, no product is observed, which implies an IC$_{50}$ value lower than 10 mM. **d** Time-dependent depletion of *trans*-β-nitrostyrene in BL21(DE3) cells transformed with either an empty pET-22b vector (EV), or a pET-22b vector containing the 4-OT gene (4-OT). In cells transformed with the 4-OT vector, *trans*-β-nitrostyrene depletion is inhibited by addition of HCHO (pink). Green: Milli-Q (MQ) water control of cells transformed with 4-OT. Errors: standard deviation of the mean ($n = 3$, each replicate in duplicate, statistical analysis: one-way ANOVA); ***$p \le 0.001$. **e** The same assay as in **d** was used to determine the extent of inhibition of 4-OT with varied HCHO concentrations; the slope between 60 and 180 min after HCHO addition was used to determine the % inhibition (EC$_{50}$ ~350 μM). Errors: standard errors of the mean ($n = 3$, each replicate in duplicate).

exceptional biologically generated carbonyl compound with respect to its reactions with biomacromolecules.

The reduced levels of bicyclic ring formation in the cases of the azetidine and piperidine analogues of the proline peptide likely reflect decreased product stability (the reactions are likely under thermodynamic control under our conditions and the pKa values of the secondary amines are similar), in particular in the case of the azetidine analogue due to ring strain in the 4,5-bicycle. It

should be noted that reactions of HCHO with N-terminal prolines are reversible and that HCHO preferentially reacts with GSH or cysteine rather than the N-terminal proline of the H2B; an observation which is consistent with the role of GSH in HCHO detoxification[9]. However, these observations do not preclude kinetically controlled reactions of locally generated HCHO, e.g. produced by demethylase catalysis, with amine and alcohol nucleophiles. H2B is reported to be methylated (at Lys37) but as yet a demethylase acting on it has not been identified[33]. It is thus possible that HCHO generated elsewhere by KDM action on histone tails, especially H3, in nucleosome complexes can react with H2B. In this regard, evidence for relatively stable hemiaminal type adducts on DNA/RNA, including as formed as a consequence of catalysis by oxygenases acting on N-methyl groups, is also of interest[3,5,34].

It is possible that the extent of reaction HCHO with N-terminal prolines, and indeed other residues, is changed under different conditions, for example under conditions of redox stress where glutathione levels are altered[35,36], or on treatment with certain electrophilic drugs capable of reacting with glutathione, for example fumarate esters used in treatment of multiple sclerosis[37]. It is also of interest that N-terminal prolines (and other residues including cysteine) are commonly modified; it is possible that such modifications, at least in part, serve to hinder reaction with HCHO and related electrophiles. HCHO also has potential to be sequestered by nucleophiles other than glutathione, including cysteine, histidine and tetrahydrofolate (THF)[1–4,38–40]. THF is of particular interest with respect to chromatin and HCHO since, unlike the JmjC KDMs including KDM4E, THF is present in the flavin-dependent lysine specific demethylase 1 (LSD1)[41] and THF reacts with HCHO to form a stable 5,10-methylene-THF cyclic adduct[40].

Although the potential functional roles, if any, of the reactions of HCHO with histones/chromatin remain to be defined, our studies using 4-OT as a model system clearly reveal the potential for non-cytotoxic levels of HCHO to inhibit specific enzyme-catalysed reactions in cells by reacting with an N-terminal prolyl residue. HCHO-mediated 4-OT inhibition was manifest despite competition with acetaldehyde and with the potential for competing reactions between HCHO and cellular nucleophiles, e.g. glutathione, cysteine, THF, and other protein nucleophiles as exemplified in a recent study on insulin[42]. The results thus highlight a question - how is it possible that HCHO and other reactive electrophiles can have specific functional effects in cells in the presence of so many nucleophiles? Indeed, the precise forms of HCHO and its adducts in cells are unknown; it seems likely that, within cells, HCHO is substantially sequestered by reactions with nucleophiles, potentially including N-terminal protein residues. The precise nature of the adducts formed by reaction of HCHO with 4-OT are unknown but the results highlight potential for modulation of enzyme and other biological macro- and small-molecule activity by reversible non-enzyme-catalysed covalent reactions with natural electrophiles.

Our ongoing work is focused on exploring the possibility of regulation of function by reactions of HCHO and other reactive carbonyl compounds with N-terminal proline and other nucleophilic protein residues (including cysteine residues), which may have important roles in enzyme catalysis or stability. Given the widespread use of HCHO as a fumigant[43], one interesting enzyme with a catalytically relevant N-terminal proline is the E. coli DNA repair enzyme formamidopyrimidine DNA glycosylase[44].

Finally, work on the functions of HCHO-producing demethylases acting on DNA, RNA and histones has to date focused on their roles in altering methylation status, the evidence for which is exceptionally strong in some cases, e.g. DNA damage repair in bacteria[45]. In other cases, the functional links between demethylation and physiology are less secure, including the demethylation reactions catalysed by oxygenases acting on histone H3 and RNA (including 6-methyladenine demethylation). The lack of clear connections between oxidation and function may simply reflect the complexities of chemical aspects of the regulation of eukaryotic gene expression. To date, however, HCHO released by demethylase catalysis has largely been regarded as a toxic byproduct. Our work suggests further work on potential physiologically relevant functions of demethylase-produced HCHO and associated hemiaminal, aminal (as described here), thiazolidine/oxazolidine and potentially more complex, including cyclic, structures is merited.

## Methods

**Peptide synthesis procedures**. See Supplementary Methods and Supplementary Fig. S36.

**Reaction conditions**. See Supplementary Methods and Supplementary Figs. S1–S5, S17–S23, S26 and S31.

**Peptide and product characterisations/methods**. See Supplementary Methods and Supplementary Figs. S1–S6, S8–S19, S21–S23, S25 and S31 for NMR and MS analyses of synthetic peptides and of reactions of peptides and histones.

**Cell based studies**. See Supplementary Methods and Supplementary Figs. S24 and S26 for details of cellular studies.

**Studies on 4-OT**. See Supplementary Methods and Supplementary Figs. S27–S30, S32–35, and S37 and Tables S2–S6 for details of studies on 4-OT and preparation of 4OT and KDM4E.

**Reporting summary**. Further information on research design is available in the Nature Portfolio Reporting Summary linked to this article.

## Data availability

The data relevant to this study are available in the main article and Supplementary Information. All data generated during the current study are available from the corresponding authors on reasonable request.

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

## Acknowledgements
T.J. was supported by the Oxford-GSK-Crick Doctoral Programme in Chemical Biology, EPSRC (EP/R512060/1) and GlaxoSmithKline. We thank the Oxford Genomics Centre at the Wellcome Centre for Human Genetics (funded by Wellcome Trust grant reference 203141/Z/16/Z) for the generation and initial processing of the sequencing data. This work was supported by funding from the Biotechnology and Biological Research Council, Cancer Research UK, the Wellcome Trust [grant no. 106244/Z/14/Z].

## Author contributions
T.J., C.J.S. and R.J.H. designed the study. T.J. conducted all experiments, except as follows: E.P. analysed the in situ HCHO and recombinant H2B samples and E.S. produced and purified KDM4E. S.S.H. analysed the 4-OT trypsin digestion and immunoprecipitated H2B digestion samples. T.J., C.J.S. and R.J.H. analysed the data and co-wrote the manuscript. All authors reviewed the manuscript.

## Competing interests
The authors declare no competing interests.
