## [Peer Review File · Communications Chemistry]

Reviewers' comments:

Reviewer #1 (Remarks to the Author):

The work "Formaldehyde Reacts With N-Terminal Proline-Residues to Give Bicyclic Aminals" By Tobias John and col addressed the potential reactivity of formaldehyde (FA) against nucleophilic groups in proteins. More specifically, they focused on N-terminal proline residues revealing the formation of an N-terminal aminal. Authors explored whether this non-enzymatic reaction could occur in histones, finding a preference towards H2B. In addition, they evaluated in vitro whether FA from a histone demethylase (KDM4) can react with the N-terminal of H2B. The work has potential and might be suitable for publication in this journal, however authors should be encouraged to work on addressing the relevance of this FA-Proline reaction and whether it might occur in cells. In general, the layout is clear, the experiments contain sufficient details, and the text is well written. Some relevant citations are missed.

Major points to be addressed before acceptance:

1- I'm not convinced about the biological relevance of this modification in H2B, and whether it can be detected in cells without exposure to FA. Authors might address this point by different approaches such as:

a. Is this H2B modification detected in this recent work?: DOI:10.1021/bc050340f. Are there proteins modified by FA that support the mechanism proposed for T. John et al regarding the reaction with proline?

b- Is there any evidence indicating that the FA-proline modification occurs in cells? Have authors performed proteomics?

2- The authors claim that histone demethylation might contribute with H2B-FA formation in cells. However, the time-lapse used in their experiments is close to 24 h. In cells, 24 h is likely a full cell cycle division. I would expect to see this reaction in a window of minutes instead of hours. How do they reconcile that the effect is seen after 24h (Fig. 1b)?

3- The concentration of FA used in each experiment should be noted in the figure legend(s). For example, in Fig.1 it is not clear how much formaldehyde is a 100 - 10000 - fold excess relative to H2B. From the methods part it could be calculated that a 10000 - fold FA to H2B would be (81uM x 10000 = 810 mM?). In the same line as my other points, would this concentration be physiologically relevant?

4- Authors might want to further discuss the biological relevance of this work. Of particular interest for this relevance are the recent works addressing the reactivity of formaldehyde with amino-acids and glutathione: <https://doi.org/10.1038/s42004-020-0324-z>; <https://doi.org/10.1038/s41467-022-28242-7>. Unfortunately, none of these works are mentioned in the text despite they are very relevant for this paper. More specifically, in the first manuscript, it is shown that in presence of cells, exogenous FA decays to a minimum within the first 12 h, forming adducts with free cysteine and histidine. In the second work, S-hydroxymethyl-glutathione was shown to be formed in cells in absence of exogenous FA, indicating cellular glutathione is a major scavenger of FA in vivo. Thus, I am not sure whether the proposed reaction between FA and H2B would be relevant in vivo, considering that FA would have to outcompete these very abundant scavengers. A further discussion might help to clarify authors' view on the relevance.

5- The first seminal work showing that histone demethylation (LSD1) produces formaldehyde is not mentioned either <https://doi.org/10.1016/j.molcel.2005.08.027>. FA generated from LSD1 was proposed to be trapped by a folate molecule found close to the active site <https://pubmed.ncbi.nlm.nih.gov/24715612/>. Thus, FA from LSD1 would require to outcompete folate in order to react with H2B. Authors addressed the competition with GSH or Cys (Fig. S20). Would folate be also a competitor of FA? Though KDM4E does not contain a folate molecule, LSD1 does.

Minor points.

1- FA was reported to induce the degradation of BRCA2 DOI: 10.1016/j.cell.2017.05.010 and more recently of POL2 (<https://doi.org/10.1038/s41586-021-04133-7>). Does any of these proteins have exposed prolines that might react with FA?

2. The reaction between FA and proline was described for FmrR, involving a cysteine. DOI: 10.1074/jbc.M116.745174. In the work evaluated, T. John and Col used the first 15 AA of H2B. The histones studied do not contain an internal cysteine residue, and thus the mechanism proposed seems to be different to that one reported for FmrR. Can authors speculate whether the full length H2B would show the same reaction with FA as the 1-15 H2B?

3- Bacterial growth analysis (instead of analyses)

4- Figure legends: Please correct nomenclature such as "1way ANOVA" Fig. 4 (one-way ANOVA).

5- Fig 4e. Please indicate the time at which these assays were performed.

6- It would be interesting to evaluate whether the FA-proline in cellular H2B can be detected in vivo (or in cells), though I understand this might not be of the scope of this work. To increase the chances of detecting very low abundant FA-protein modifications in cells, authors might perform mass spectrometry (SILAC etc) in cells grown in presence of the glutathione synthesis inhibitor L-buthionine sulfoximine (L-BSO) <https://doi.org/10.1038/s41467-022-28242-7>, or of the ADH5 inhibitor N6022 <https://doi.org/10.1038/s41467-020-20754-4>. Blocking GSH synthesis and ADH5 will cause a drop in GSH and an increase in free FA, thus it would be more likely to detect these low-abundant FA-protein modifications.

Reviewer #2 (Remarks to the Author):

This study explores the intriguing possibility of formaldehyde acting as regulatory molecule via modification of N-terminal proline residues in proteins. The number of proteins with N-terminal prolines is predicted to be above 200, thus this form of protein modification could lead to significant biological effects. The authors show convincing evidence that formaldehyde reacts specifically with N-terminal proline peptides to form a bicyclic product using different human histone peptides. Although identifying HCOH modification of H2B in HEK293 cells was not successful, the authors do show formaldehyde treatment inhibits the enzyme 4-OT in viable bacterial cells. Overall, the manuscript presents new evidence for formaldehyde reacting with N-terminal proline residues. These findings are of broad interest and may have significant biological consequences.

Some comments for the authors to address:

1. Is it known whether KDM4E binds to the N-terminus of the H2B 15mer? Would this facilitate the reaction of H2B with HCHO?

2. In the reaction scheme shown in Figure S7, the protons are not balanced in the first step. Consider omitting the lone proton on the left-hand side and the possibility that after the nucleophilic attack (or coinciding with the attack) the proton from the proline nitrogen is transferred to the oxygen of HCHO to give the hemiaminal species.
3. If you compare the chemical properties of the nitrogen in proline and the proline analogues (azetidine and piperidine), are any trends in reactivity with HCHO related to differences in pKa values?
4. Is the unstability of the product with acetaldehyde due to steric hinderance of the methyl group during the cyclization reaction? Is this why no reaction products are observed with the bulkier aldehyde groups such as pyruvate and acetone?
5. Did the authors obtain an inhibition constant for HCOH inhibition of 4-OT? If so, does HCOH behave as a irreversible or reversible inhibitor?
6. The authors discuss that GSH and cysteine are scavengers of formaldehyde. It would be helpful for the authors to comment on whether HCOH modification of N-terminal proline residues may be of more consequence during oxidative stress conditions or nutritional environments that limit GSH synthesis.
7. Fig S26, panel C- correct GAPHD

Responses to specific comments from the referees are given below. We thank the referees for their helpful comments, which we address below and which we think have led to a much improved manuscript. Overall, we appreciate their interest in the new reaction identified. We also appreciate that the relevance of our (bio)chemical studies to physiology is of interest but remains to be validated, as is the case for multiple post-translational modifications, including on histones. It was not our intention to claim physiological relevance – we make this very clear in the revised manuscript – but to promote and enable interdisciplinary work in the field, including from chemists to define the precise nature of the reactions of HCHO and related compounds with biomolecules – a field that has not been a recent priority. We are also keen to highlight the (unanswered) basic science question of how is it possible that HCHO and other reactive electrophiles can have functional effects in cells in the presence of so many nucleophiles? The work with the 4-OT model system reveals the potential of the new reaction to alter function in cells.

Responses to comments from reviewer 1:

I'm not convinced about the biological relevance of this modification in H2B, and whether it can be detected in cells without exposure to FA. Authors might address this point by different approaches such as:

a. Is this H2B modification detected in this recent work?: DOI:10.1021/bc050340f. Are there proteins modified by FA that support the mechanism proposed for T. John et al regarding the reaction with proline?

Thanks for bringing this study to our attention – it focuses on insulin, which does not contain an N-terminal proline residue, so the work is not directly relevant; however, it nicely exemplifies the complexity of reactions of HCHO with proteins and we added a citation and commented on it in the discussion of the revised manuscript.

Is there any evidence indicating that the FA-proline modification occurs in cells? Have authors performed proteomics?

Despite extensive efforts, to date our proteomics analyses have not provided coverage of the N-terminal region of H2B, meaning we were unable to confirm formation of a proline- and formaldehyde-derived imidazolidinone (SI Figure S25 in the revised manuscript). To our knowledge, no proline- and formaldehyde-derived imidazolidinones have been reported on histone proteins, although imidazolidinone formation has been proposed on many proteins after aldehyde exposure. This is particularly pertinent in the case of haemoglobin, for which N-terminal formaldehyde- and acetaldehyde-derived imidazolidinones have been reported in human samples, including at endogenous aldehyde levels. The apparent lack of observation of proline- and formaldehyde-derived imidazolidinones reported may reflect (i) the substantial technical challenge of identifying reversible protein N-terminal-derived peptides during proteomics analyses (as exemplified by our work), (ii) the relative paucity of proteins containing unmodified proline residues on their N-termini, and (iii) the incredible complexity of the biochemistry involved, in particular with respect to post-translational modifications of histone tails. As described below in response the 4th point raised by the reviewer, this is exactly why we moved to study (potential) HCHO-mediated regulation in a much simpler system, that is catalysis by a single enzyme that reacts with HCHO. Perhaps the most important results in the manuscript concern our studies using 4-OT as a model system that clearly reveal the potential for HCHO to have a functional effect in cells.

The authors claim that histone demethylation might contribute with H2B-FA formation in cells. However, the time-lapse used in their experiments is close to 24 h. In cells, 24 h is likely a full cell cycle division. I would expect to see this reaction in a window of minutes instead of hours. How do they reconcile that the effect is seen after 24h (Fig. 1b)?

Although the reaction between the proline residue and formaldehyde is slow under our conditions, even a small global level of imidazolidinone could have a marked effect on H2B-dependent epigenetic regulation at the level of individual genes. There is also potential for high localised formaldehyde concentrations near chromatin given the action of formaldehyde-producing demethylases on methylated histones and DNA. Other local environmental conditions (e.g. solvent viscosity, protein structure) will also affect the rate of reaction. We have modified the discussion to emphasise these points. As described above, the complexity of eukaryotic transcription is why we explored the potential of the new post-translational modification with a much simpler system (4-OT).

The concentration of FA used in each experiment should be noted in the figure legend(s). For example, in Fig.1 it is not clear how much formaldehyde is a 100 - 10000 - fold excess relative to H2B. From the methods part it could be calculated that a 10000 - fold FA to H2B would be (81uM x 10000 = 810 mM?). In the same line as my other points, would this concentration be physiologically relevant?

We have added the concentrations of formaldehyde used in the relevant figure legends. We agree that the higher concentrations are unlikely to be physiologically relevant but these experiments demonstrate the concentration dependence of adduct formation and enabled product characterisation. Of course local concentrations in vivo could be high, but there is no analytical method to measure these. We have included a sentence on this in the discussion.

Authors might want to further discuss the biological relevance of this work. Of particular interest for this relevance are the recent works addressing the reactivity of formaldehyde with amino-acids and glutathione: <https://doi.org/10.1038/s42004-020-0324-z>; <https://doi.org/10.1038/s41467-022-28242-7>. Unfortunately, none of these works are mentioned in the text despite they are very relevant for this paper. More specifically, in the first manuscript, it is shown that in presence of cells, exogenous FA decays to a minimum within the first 12 h, forming adducts with free cysteine and histidine. In the second work, S-hydroxymethyl-glutathione was shown to be formed in cells in absence of exogenous FA, indicating cellular glutathione is a major scavenger of FA in vivo. Thus, I am not sure whether the proposed reaction between FA and H2B would be relevant in vivo, considering that FA would have to outcompete these very abundant scavengers. A further discussion might help to clarify authors' view on the relevance.

We thank the reviewer for these comments – the reviewer is entirely correct regarding the references and we have cited these – apologies. The reviewer highlights the key basic science question – how is it possible that HCHO and other reactive electrophiles can have functional effects in cells in the presence of so many nucleophiles? With hindsight this could have been clearer in the discussion and it is made so in the revised manuscript. Reaction of H2B with HCHO and indeed any other reaction of HCHO – functionally relevant or not – will have to compete with reactions with glutathione and other cellular nucleophiles, as demonstrated by our competition experiments (Figures S20-S22) and the literature, as pointed out by the reviewer. In the

revised discussion we state the possibility of locally generated HCHO reacting with chromatin. We completely agree that the functional roles, if any, of the reactions of HCHO with histones / chromatin remain to be defined. We expect that the new reaction reported in our biochemical work will promote and enable research by others to study this, as has been the case with our work on the incredibly complex roles of histone modifications, as exemplified by recent studies on N-methyl arginine demethylation. The complexity of the biochemistry of eukaryotic chromatin is precisely why we moved to study (potential) HCHO-mediated regulation in a massively simpler system, that is catalysis by a single enzyme that reacts with HCHO. Perhaps the most important result in the manuscript concerns our studies using 4-OT as a model system that clearly reveal the potential for non-cytotoxic levels of HCHO to inhibit specific enzyme-catalysed reactions in cells by reacting with an N-terminal prolyl residue – this is despite all the issues regarding competition, as appreciated by and raised by the reviewer. We agree there is much still to be learned about HCHO chemistry in cells, which is difficult to study (in particular for freely reversible reactions that are not suitable for analysis by techniques such as metabolomics), but this result clearly reveals the potential for functional relevance for the novel reaction described in our work.

The first seminal work showing that histone demethylation (LSD1) produces formaldehyde is not mentioned either <https://doi.org/10.1016/j.molcel.2005.08.027>. FA generated from LSD1 was proposed to be trapped by a folate molecule found close to the active site <https://pubmed.ncbi.nlm.nih.gov/24715612/>. Thus, FA from LSD1 would require to outcompete folate in order to react with H2B. Authors addressed the competition with GSH or Cys (Fig. S20). Would folate be also a competitor of FA? Though KDM4E does not contain a folate molecule, LSD1 does.

Thank you for these very helpful comments and we apologise for the incomplete citations – we have added these and other relevant references (including to the pioneering work of Paik and Kim (DOI: 10.1016/0006-291x(73)91383-1, showing HCHO is a product of histone demethylation) in the revised manuscript. We have also added a comment regarding THF including studies on competition with glutathione for reaction with HCHO.

Minor points

FA was reported to induce the degradation of BRCA2 DOI: 10.1016/j.cell.2017.05.010 and more recently of POL2 (<https://doi.org/10.1038/s41586-021-04133-7>). Does any of these proteins have exposed prolines that might react with FA?

A major objective of our chemistry focussed manuscript is to promote work by others on the reaction of N-terminal residues with aldehydes, in particular in the context of disease. We thus agree looking for disease-relevant proteins with N-terminal prolines that may react with aldehydes to give cyclic structures is of considerable interest. Neither of the proteins identified by the reviewer is reported to contain a proline on their N-terminus. In the revised manuscript we do comment on one such example with a catalytically relevant N-terminal proline, that is the *E. coli* DNA repair enzyme formamidopyrimidine DNA glycosylase. Given the links between HCHO and impaired DNA damage repair, this protein is of particular interest.

The reaction between FA and proline was described for FmrR, involving a cysteine. DOI: 10.1074/jbc.M116.745174. In the work evaluated, T. john and col used the first 15 AA of H2B. The histones studied do not contain an internal cysteine residue, and thus the mechanism proposed seems to be different to

that one reported for FmrR. Can authors speculate whether the full length H2B would show the same reaction with FA as the 1-15 H2B?

Thank you for this interesting idea. H2B does not have a cysteine residue and so cannot react via a similar mechanism. We have added a comment on this in the discussion. We do appreciate that reactions with full length histones / nucleosomes may be more complex and have also commented on this.

Bacterial growth analysis (instead of analyses)

Thank you - this has been corrected.

Figure legends: Please correct nomenclature such as "1way ANOVA" Fig. 4 (one-way ANOVA).

Thank you - this has been corrected.

Fig 4e. Please indicate the time at which these assays were performed.

The information has been added to the Figure 4 legend.

It would be interesting to evaluate whether the FA-proline in cellular H2B can be detected in vivo (or in cells), though I understand this might not be of the scope of this work. To increase the chances of detecting very low abundant FA-protein modifications in cells, authors might perform mass spectrometry (SILAC etc) in cells grown in presence of the glutathione synthesis inhibitor L-buthionine sulfoximine (L-BSO) <https://doi.org/10.1038/s41467-022-28242-7>, or of the ADH5 inhibitor N6022 <https://doi.org/10.1038/s41467-020-20754-4>. Blocking GSH synthesis and ADH5 will cause a drop in GSH and an increase in free FA, thus it would be more likely to detect these low-abundant FA-protein modifications.

Thank you for these very helpful suggestions, which we will pursue in future biological work. We have commented on the possibility that drug treatment may alter HCHO / glutathione levels in the revised discussion.

Responses to comments from reviewer 2:

Is it known whether KDM4E binds to the N-terminus of the H2B 15mer? Would this facilitate the reaction of H2B with HCHO?

KDM4 demethylases are not reported to bind the N-terminus H2B, although we cannot rule out interactions. We have noted this in the discussion.

In the reaction scheme shown in Figure S7, the protons are not balanced in the first step. Consider omitting the lone proton on the left-hand side and the possibility that after the nucleophilic attack (or coinciding with the attack) the proton from the proline nitrogen is transferred to the oxygen of HCHO to give the hemiaminal species.

We have modified the reaction scheme accordingly.

If you compare the chemical properties of the nitrogen in proline and the proline analogues (azetidine and piperidine), are any trends in reactivity with HCHO related to differences in pKa values?

There is no clear trend between the reactivity and relevant pKa values of the secondary amines tested, which are all similar (e.g. for proline 10.60 versus 10.68 for azetidine carboxylate). Instead it is likely that the stability

of the bicyclic ring systems is a major factor, in particular for the strained 4,5-ring system products of the azetidine analogue. The reactions are reversible and under thermodynamic control – we have made this clear in the revised manuscript.

Is the unstability of the product with acetaldehyde due to steric hinderance of the methyl group during the cyclization reaction? Is this why no reaction products are observed with the bulkier aldehyde groups such as pyruvate and acetone?

The different reaction profiles observed for acetaldehyde and formaldehyde likely reflect, at least in part, differences in steric factors. It should be noted that all the reactions are reversible, suggesting thermodynamic rather than kinetic control. We have commented on these in the revised discussion in the main text.

Did the authors obtain an inhibition constant for HCOH inhibition of 4-OT? If so, does HCOH behave as a irreversible or reversible inhibitor?

We have added all the information in Figure 4. The reaction is reversible, as indicated above.

The authors discuss that GSH and cysteine are scavengers of formaldehyde. It would be helpful for the authors to comment on whether HCOH modification of N-terminal proline residues may be of more consequence during oxidative stress conditions or nutritional environments that limit GSH synthesis.

Thank you – this is a good point and has been commented on in the revised discussion, including by addition of relevant references. This idea is also relevant with respect to treatment with certain electrophilic drugs capable of reacting with glutathione, for example fumarate esters used in treatment of multiple sclerosis. We have also commented on this.

Fig S26, panel C- correct GAPHD

Thank you - this has been corrected.

REVIEWERS' COMMENTS:

Reviewer #1 (Remarks to the Author):

The authors have provided clarifications to all my queries. This insightful work should be published in its current form.